# Diffusion-Based Offline RL for Improved Decision-Making in Augmented ARC Task

## Abstract

Effective long-term strategies enable AI systems to navigate complex environments by making sequential decisions over extended horizons. Similarly, reinforcement learning (RL) agents optimize decisions across sequences to maximize rewards, even without immediate feedback. To verify that Latent Diffusion-Constrained Q-learning (LDCQ), a prominent diffusion-based offline RL method, demonstrates strong reasoning abilities in multi-step decision-making, we aimed to evaluate its performance on the Abstraction and Reasoning Corpus (ARC). However, applying offline RL methodologies to enhance strategic reasoning in AI for solving tasks in ARC is challenging due to the lack of sufficient experience data in the ARC training set. To address this limitation, we introduce an augmented offline RL dataset for ARC, called Synthesized Offline Learning Data for Abstraction and Reasoning (SOLAR), along with the SOLAR-Generator, which generates diverse trajectory data based on predefined rules. SOLAR enables the application of offline RL methods by offering sufficient experience data. We synthesized SOLAR for a simple task and used it to train an agent with the LDCQ method. Our experiments demonstrate the effectiveness of the offline RL approach on a simple ARC task, showing the agent's ability to make multi-step sequential decisions and correctly identify answer states. These results highlight the potential of the offline RL approach to enhance AI's strategic reasoning capabilities.

## 1 Introduction

Effective long-term strategies involve deliberate reasoning, which refers to the thoughtful evaluation of options to determine the best course of action (Kahneman, 2011). This type of reasoning requires conscious effort and allows intelligent beings to systematically plan and execute multi-step strategies to achieve complex long-term goals. Similarly, reinforcement learning (RL) agents make decisions with the goal of maximizing rewards over extended sequences of actions, even without immediate feedback. In both cases, reasoning involves considering a sequence of actions to reach an optimal outcome. We believe that the way Q-values guide an RL agent toward desired outcomes aligns with the subgoals of deliberate reasoning, particularly in terms of multi-step decision-making to achieve long-term objectives.

Recent approaches to offline RL combined with generative diffusion models have shown significant improvements in multi-step strategic decision-making abilities for future outcomes (Janner et al., 2022; Ajay et al., 2023; Liang et al., 2023; Li et al., 2023). In particular, Latent Diffusion-Constrained Q-learning (LDCQ) (Venkatraman et al., 2024) leverages diffusion models to sample various latents that compress multi-step trajectories. These latents are then used to guide the Q-learning process. By generating diverse data based on in-distribution samples, diffusion models help overcome the limitations of fixed datasets. This integration of diffusion models into offline RL enhances agents' reasoning abilities, allowing them to consider multiple plausible trajectories across extended sequences.

We aim to apply the offline RL method to tackle reasoning benchmarks that demand advanced reasoning capabilities. To this end, we chose the Abstraction and Reasoning Corpus (ARC) (Chollet, 2019), one of the key benchmarks for measuring the abstract reasoning ability in AI. As shown in Figure 1, the ARC training set consists of 400 grid-based tasks, each requiring the identification of common rules from demonstration examples, which are then applied to solve the test examples. ARC tasks are particularly challenging for AI models because they demand high-level reasoning abilities,

integrating core knowledge priors such as objectness, basic geometry, and topology (Chollet, 2019). These core knowledge priors guide the decision-making process for selecting the appropriate actions. Therefore, we believe that agents trained with offline RL methods can leverage these core knowledge priors by learning from the experienced data.

However, the existing ARC training dataset lacks sufficient trajectories for training agents with offline RL methods. To address this limitation, we propose Synthesized Offline Learning data for Abstraction and Reasoning (SOLAR), a dataset for training offline RL agents. SOLAR provides diverse trajectory data, allowing the agent to encounter various actions shaped by the core knowledge priors across different episodes. In this research, we synthesized SOLAR for a simple task using the SOLAR-Generator, which creates data according to the desired conditions. The synthesized SOLAR was then used to train agents using the LDCQ method. Through training with LDCQ on SOLAR, agents demonstrated the ability to devise pathways to correct answer states, even generating solution paths not present in the training data. This approach highlights the potential of diffusion-based offline RL methods to enhance AI's reasoning capabilities.

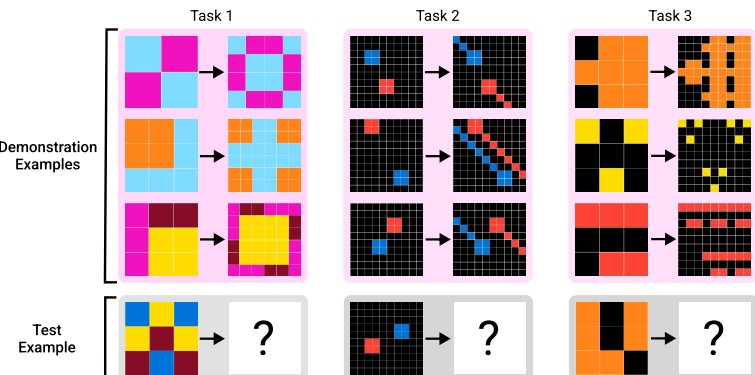

Figure 1: Three tasks in ARC. Each task consists of demonstration examples and a test example. Each example has an input grid and an output answer grid. Each pixel in the grid is matched to a color corresponding to a value in the range 0–9. ARC requires identifying common rules from the demonstration examples and applying them to solve the test example correctly. Despite recent advancements in AI, current models have consistently underperformed compared to humans on the ARC benchmark (Chollet et al., 2024; Johnson et al., 2021).

## 2 PRELIMINARIES

### 2.1 ARC LEARNING ENVIRONMENT (ARCLE)

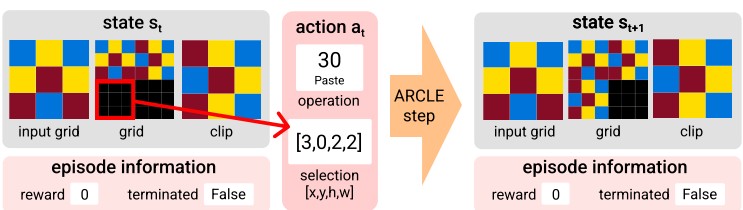

Figure 2: An example of a single step in ARCLE. In this example step, the action has an operation 30 (`Paste`) and a selection of $[3, 0, 2, 2]$. The top-left coordinate of the selection box is $[3, 0]$ and the bottom-right coordinate is $[5, 2]$. $[h_t, w_t]$ is calculated by subtracting $[3, 0]$ from $[5, 2]$. When ARCLE executes this action, the current clipboard is pasted into the bounding box specified by the selection on the current grid. It then returns episode information, including the reward and termination status.

ARCLE (Lee et al., 2024) is a Gymnasium-based environment developed to facilitate RL approaches for solving ARC tasks. ARCLE frames ARC tasks within a Markov Decision Process (MDP) structure, providing an environment where agents can interact with and manipulate grid-based tasks. This MDP structure enables ARC tasks to be solved through sequential decision-making.

ARCLE handles states and actions following the O2ARC web interface (Shim et al., 2024). As shown in Figure 2, when ARCLE executes an action $a_t$ on the current state $s_t$, it returns the next state $s_{t+1}$, along with episode information about the reward and termination status. A state $s_t$ consists of (input grid, $\text{grid}_t$, $\text{clipboard}_t$) at timestep $t$. The input grid represents the initial state of the test example, the $\text{grid}_t$ denotes the current grid at time $t$ after several actions have been applied, and the $\text{clipboard}_t$ stores the copied grid by the Copy operation. An action $a_t$ consists of ($\text{operation}_t, x_t, y_t, h_t, w_t$), where $\text{operation}_t$ represents the type of transformation, $x_t$ and $y_t$ denote the coordinates of the top-left point of the selection box, and $h_t$ and $w_t$ represent the difference between the bottom-right and top-left coordinates. All subsequent notations for $s_t$ and $a_t$ will adhere to this definition for clarity. Reward is only given when the Submit operation is executed at the answer state, and the episode terminates either after receiving the reward or when Submit is executed across multiple trials. All possible operations are mentioned in Appendix B.1.

## 2.2 DIFFUSION-BASED OFFLINE REINFORCEMENT LEARNING

Offline RL focuses on learning policies from previously collected data, without interacting with the environment. However, Offline RL faces challenges, including data distribution shifts, limited diversity in the collected data, and the risk of overfitting to biased or insufficiently representative samples. To address these issues, several works in offline RL have focused on improving learning efficiency with large datasets and enhancing generalization to unseen scenarios while balancing diversity and ensuring data quality (Fujimoto et al., 2019; Kidambi et al., 2020; Levine et al., 2020).

Recent offline RL methods offer promising solutions in complex tasks with diverse samples through diffusion models. For instance, Diffuser (Janner et al., 2022) generates trajectories by learning trajectory distributions, reducing compounding errors. In addition, diffusion-based offline RL approaches, such as Decision Diffuser (DD) (Ajay et al., 2023), Latent Diffusion-Constrained Q-learning (LDCQ) (Venkatraman et al., 2024), AdaptDiffuser (Liang et al., 2023), and HDMI (Li et al., 2023), have demonstrated the effectiveness of combining diffusion models with offline RL.

## 3 SYNTHESIZED OFFLINE LEARNING DATA FOR ABSTRACTION AND REASONING (SOLAR)

We developed a new dataset called Synthesized Offline Learning Data for Abstraction and Reasoning (SOLAR) that can be used to train offline RL methods. Solving ARC tasks can be considered a process of making multi-step decisions to transform the input grid into the output answer grid. We believe that the process of making these decisions inherently involves applying core knowledge priors such as objectness, goal-directedness, numbers and counting, and basic geometry and topology (Chollet, 2019), which are necessary for solving ARC tasks. The ARC training set lacks information on how to solve the tasks, providing only a set of demonstration examples and a test example for each task, as shown in Figure 1. To address this, we aim to provide the trajectory data to solve the tasks through SOLAR, enabling learning of how actions change the state based on the application of core knowledge priors. With SOLAR-Generator, we can generated new input-output grid pairs that adhere to the defined rules and created trajectory data to solve problems based on the newly generated data.

### 3.1 SOLAR STRUCTURE

As shown in Figure 3, SOLAR consists of two key components: *Demonstration Examples* and *Test Example with Trajectory*. The demonstration examples and the test examples serve the same roles as in ARC. Through the demonstration examples, the common rule for transforming the input grid to the output grid is identified and then applied to solve the test example. Trajectory data means the episode data that starts from test input $s_0$.

SOLAR contains various transition data ($s_t, a_t, s_{t+1}, r_t$), where actions $a_t$ are taken in different states $s_t$, then the result $s_{t+1}$ observed and the reward $r_t$ is given. To facilitate effective learning and a combination of core knowledge, we use ARCLE (Lee et al., 2024). When designing the reward system based on ARCLE, it is crucial for the agent to recognize when it has reached the answer state. Successfully identifying the answer state implies that the agent has understood the underlying analogy and executed the necessary ARCLE actions to arrive at the correct solution. This recognition is critical as it demonstrates the agent's comprehension of the task's inherent logic and its ability to

apply appropriate problem-solving strategies. In ARCLE, the reward is given only when the agent correctly predicts the `Submit` operation and the submitted grid matches the answer grid. Therefore, every SOLAR episode concludes with the `Submit` operation, where the agent submits an answer and determines whether it is correct.

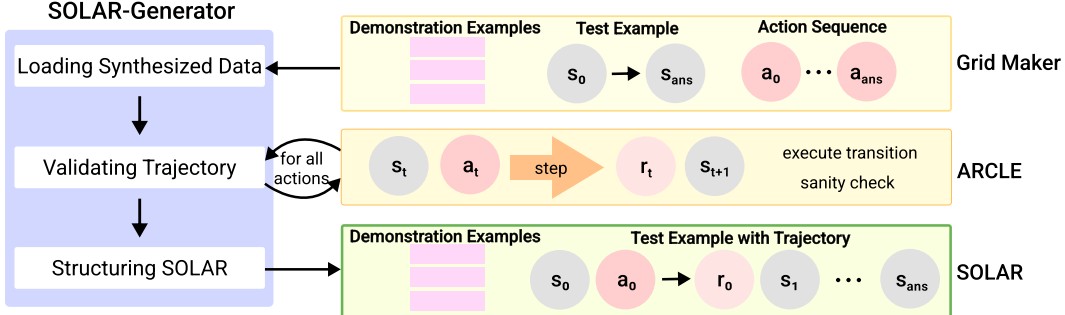

Figure 3: Data synthesis procedure with SOLAR-Generator. The state and actions consist of as mentioned in Section 2.1. 1) Loading Synthesized Data: The Grid Maker module applies constraints, augments input-output pairs, and synthesizes solutions for specific tasks by utilizing actions. 2) Validating Trajectories: Checks whether the generated actions are executable in ARCLE. 3) Structuring SOLAR: Organizes and stores the synthesized data in SOLAR based on the defined format. This step determines what information to include in the dataset and whether to segment episodes into fixed-length chunks or store them as a whole.

## 3.2 SOLAR-GENERATOR

We developed the SOLAR-Generator to synthesize SOLAR. SOLAR-Generator augments ARC trajectories by following ARCLE formalism, addressing the inherent complexity and diversity of ARC tasks. Figure 3 illustrates the data synthesis procedure, which is carried out in three steps: 1) Loading Synthesized Data, 2) Validating Trajectories with ARCLE, and 3) Structuring SOLAR.

**Loading Synthesized Data**   To create trajectories, we generate demonstration examples, test examples, and action sequences (selections and operations). These data are synthesized through the Grid Maker, which generates them according to the given rules. Each task has its own specific Grid Maker that defines the task's constraints and rules, using common parameters such as the maximum grid size and the number of demonstration examples per test example. By defining the rules for generating data, the user can adjust the difficulty level and other specific details. At this stage, the Grid Maker synthesizes only grid pairs and possible action sequence. The full trajectory data for the test example is constructed after validating through ARCLE. More details about how the Grid Maker synthesizes the input-output grids and action sequences are described in Appendix B.

**Validating Trajectories with ARCLE**   After synthesizing various grids and action sequences with the Grid Maker, the SOLAR-Generator checks whether the action sequences are valid in ARCLE. The Grid Maker serves as a data loader, enabling it to load and validate the synthesized data. ARCLE utilizes this data loader to load the synthesized input-output pairs, perform actions at the $t$-th state $s_t$ of the test example, and provide intermediate states, rewards, and termination status for each step, verifying that each action is correctly executed in the current state. This step is particularly important for non-optimal trajectories, where operations and selections may be generated randomly, as invalid selections can sometimes be synthesized by the Grid Maker. For *gold standard* trajectories, intended as correct solutions, SOLAR-Generator ensures that the final grid of the trajectory matches the expected output grid of the test example. As a result, this stage is useful for checking and debugging the synthesized trajectories, preventing unintended errors.

**Structuring SOLAR**   After the trajectory validation is complete, the episodes are saved into SOLAR. In this step, user can determine the necessary information to include in SOLAR. At its core, SOLAR includes episodes consisting of state, action, reward, and termination information at each step, which are essential for training with offline RL methods. In addition to the previously mentioned

information, SOLAR can also store various data from ARCLE, such as grid sizes at each step, binary mask versions of selections, and other relevant information needed for different learning methods. In this research, we designed the data to work with methods like LDCQ, which require trajectories of fixed horizon length $H$. Therefore, the trajectories are segmented into fixed-length chunks with a horizon length of $H$.

Through these three steps, SOLAR-Generator synthesizes diverse solutions by altering action orders or using alternative operation combinations. This is achieved by the Grid Maker, which generates data using pre-implemented algorithms, enabling the user to create as many trajectories as needed. SOLAR provides a sufficient training set for learning various problem-solving strategies. By offering diverse trajectories while adhering to the task-solving criteria, SOLAR bridges the gap between ARC's reasoning challenges and the sequential decision-making process of offline RL. The whole algorithm for SOLAR-Generator is described in Algorithm 1. For additional details about SOLAR and SOLAR-Generator, see Appendix B.

---

**Algorithm 1:** SOLAR-Generator

1 Input: task set $T$, maximum grid size $(H,W)$, number of samples $N$, number of examples $E$

2 **for** $task \in T$ **do**
3    # Load the synthesized data $\mathcal{D}_s$ from the Grid Maker for the *task*
4    $\mathcal{D}_s \leftarrow$ Grid Maker(*task*, $(H, W)$, $N$, $E$)
5    **for** $data \in \mathcal{D}_s$ **do**
6       # Extract the demonstration examples, test example, and actions for each episode
7       *trajectory_ID, dem_ex, input_grid, output_grid, operations, selections* $\leftarrow$ *data*
8       Add *trajectory_ID, dem_ex, input_grid, output_grid* to episode $\tau_{data}$
9       # Set the initial state
10      $current\_grid_0 \leftarrow input\_grid$
11      $clip\_grid_0 \leftarrow None$
12      $t \leftarrow 0$
13      $s_t \leftarrow (input\_grid, current\_grid_0, clip\_grid_0)$
14      **for** $(opr_t, sel_t) \in (operations, selections)$ **do**
15         $a_t \leftarrow (opr_t, sel_t)$
16         **if** $a_t$ can be performed in $s_t$ **then**
17            # Update state and episode information using ARCLE
18            $current\_grid_{t+1}, clip\_grid_{t+1}, reward_t, terminated_t \leftarrow \texttt{ARCLE.step}(s_t, a_t)$
19            Add $s_t, a_t, reward_t, terminated_t$ to $\tau_{data}$
20            $s_{t+1} \leftarrow (input\_grid, current\_grid_{t+1}, clip\_grid_{t+1})$
21            $t \leftarrow t + 1$
22         **end**
23         **else**
24            Save wrong data for debugging
25            break
26         **end**
27      **end**
28      **if** *"gold-standard"* in *trajectory_ID* and *current_grid* $\neq$ *output_grid* **then**
29         Save wrong data for debugging
30      **end**
31      **else**
32         Save episode $\tau_{data}$
33      **end**
34    **end**
35 **end**

---

## 4 SOLAR FOR A SIMPLE TASK

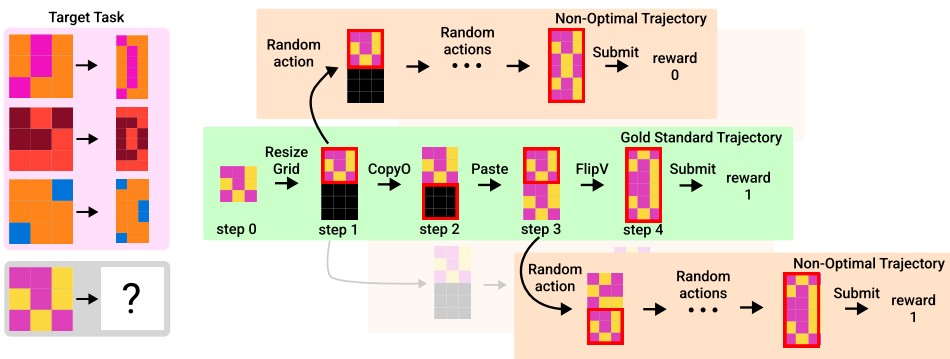

Figure 4: SOLAR episodes for a simple task. It consist of three demonstration examples (pink box) and a test example (gray box) with a trajectory (green or orange box). The gold standard episode contains the steps to solve the problem by using the core knowledge priors properly. The non-optimal episodes branch off at a random step within the standard trajectory, performing random operations such as `Rotate`, `Flip`, or `Copy` & `Paste`, and then `Submit` after a certain number of steps.

To evaluate whether an RL agent can correctly identify and submit at the answer state, even when non-optimal trajectories are included in the training dataset, we included episodes where the `Submit` operation is conducted in non-answer states. Our experimental objectives are: 1) To assess whether the model can reach the answer state when non-optimal trajectories are mixed with gold standard trajectories, and 2) To determine whether the model can recognize the answer state and perform the `Submit` action correctly.

We designed *Simple Task* that requires core knowledge priors such as objectness and geometry. This task necessitates the ability to consider the input grid as an object and then perform actions based on this object. We constrained the maximum grid size to 10x10, and each episode includes three demonstration pairs. In constructing SOLAR for this task, the dataset includes both gold standard episodes—which successfully reach the answer state and perform the `Submit` action—and non-optimal episodes—which follow random paths that may or may not reach the answer state. The inclusion of non-optimal trajectories aimed to evaluate whether the agent could recognize the answer state and perform the `Submit` action appropriately, thereby assessing its reasoning abilities rather than simply mimicking the dataset actions.

As explained in Section 3, the Grid Maker first generates input-output pairs for several demonstration examples and test examples. For each input-output pair, both gold-standard and non-optimal trajectories are generated. In the gold standard episode for this task, as shown in Figure 4, the steps are as follows: 1) `ResizeGrid` to make the grid two times longer vertically, 2) `CopyO` to copy the upper half of the current grid, as it matches the input grid, 3) `Paste` to apply it to the lower half of the grid, 4) `FlipV` to vertically flip the upper half of the current grid, and 5) `Submit`, as it reaches the answer state.

In the non-optimal episodes, the trajectories initially follow the gold standard trajectory but deviate at a random step to execute random actions for eight steps. We constrained the random operations to `FlipV` (vertical flip), `FlipH` (horizontal flip), `Rotate90` (counterclockwise rotation), `Rotate270` (clockwise rotation), and `CopyO` (copy the selected area to clipboard) & `Paste`. For selection, it was constrained to either two options (upper half or lower half of the current grid) or three options (upper half, lower half, or the whole grid). Specifically, there are two options for `Rotate90`, `Rotate270`, and `CopyO`, and three options for the others. This simplified selection allows for focusing on assessing the AI's decision-making by sequentially combining operations.

For each test example, one gold-standard episode and nine non-optimal episodes were generated. In total, 500 test examples and 5,000 episodes are generated. Consequently, the training set was composed such that approximately 10% of the total episodes included the `Submit` operation at the correct answer state.

## 5 EXPERIMENTS AND RESULTS

Using the training dataset for a simple task, we trained an agent with Latent Diffusion-Constrained Q-learning (LDCQ) (Venkatraman et al., 2024), a prominent diffusion-based offline RL method. LDCQ utilizes diffusion models to sample diverse latent representations that encapsulate multi-step trajectories, enabling the Q-learning process to effectively explore various possible action sequences. More details about training process with LDCQ are described in Appendix A.

### 5.1 EVALUATION PROCESS USING ARCLE

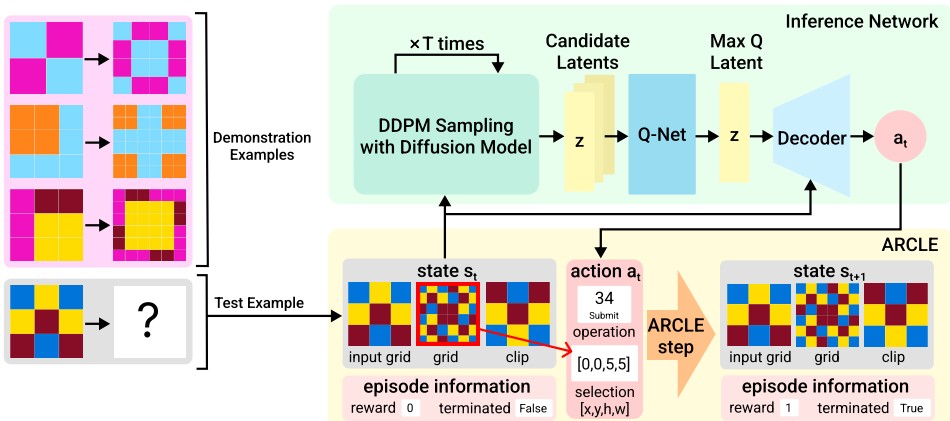

Figure 5: Inference framework for solving ARC tasks. ARCLE loads the task from the dataset and manages state information as well as the termination status of the current evaluation episode. The inference network of LDCQ performs DDPM sampling on the given state to extract candidate latents, then decodes the corresponding action for max Q latent, and sends it to ARCLE. ARCLE executes the action and updates the state information accordingly. This process alternates between ARCLE and the inference network, continuing the inference until the episode ends.

After training the agent using LDCQ on the SOLAR dataset, we conducted an evaluation of its performance. To evaluate our experiment, we synthesized an evaluation SOLAR set with 100 test examples, each paired with three synthesized demonstration examples. To measure the effectiveness of decision-making using the Q-function, two accuracy metrics are measured: 1) Whether the agent reaches the answer state, and 2) Whether it predicts the `Submit` operation at the answer state to receive a reward.

The evaluation process is carried out through ARCLE, which manages the problem and its corresponding solution from SOLAR. ARCLE handles state transitions, performs actions, and verifies whether the submitted solution is correct. As depicted in Figure 5, ARCLE interacts with the LDCQ inference network by alternating the exchange of $s_t$ and $a_t$, facilitating the decision-making process toward reaching the correct answer state. The latent $z_t$ represents a segment trajectory spanning from timestep $t$ to $t + H - 1$, and is trained to accurately decode actions for any state within this segment trajectory.

In the original LDCQ methodology, inference is performed by executing several horizons using a single latent, followed by predicting the next latent. However, in the task used for this research, which has a gold standard trajectory consisting of five steps, it is possible to complete the task with just one latent sampling from the initial state. While reaching the correct answer in this manner is not inherently problematic, one of the primary goals of this research is to analyze whether the agent learns the knowledge prior to how actions work across various states. Thus, instead of focusing solely on solving the problem in as few steps as possible, only one action is conducted per latent. With this, the results demonstrate that the agent can make far-sighted decisions to reach the answer not just from the beginning to the end, but also through intermediate steps.

## 5.2 RESULTS

To demonstrate the strengths of the diffusion-based offline RL method guided by Q-function, we compare three approaches:

- **VAE prior (VAE)**: This method uses a latent sampled from the VAE state prior $p_\omega(z_t|s_t)$. The VAE state prior is trained in $\beta$-VAE training stage by calculating the KL divergence between $p_\omega(z_t|s_t)$ and the posterior $q_\phi(z_t|\tau_t)$, aligning the latent distribution with the trajectory starting from state $s_t$.

- **Diffusion prior (DDPM)**: This method uses a latent sampled from the diffusion model $p_\psi(z_t|s_t)$ through the DDPM method (Ho et al., 2020). The sampled latents closely resemble the training data, with added variance during the denoising process. This method is similar to behavior cloning in that it operates without guidance from rewards or value functions.

- **Max Q latent (LDCQ)**: This method selects a latent with the highest Q-value from those sampled by the diffusion model, $\mathrm{argmax}_{z \sim p_\psi(z_t|s_t)} Q(s_t, z)$, to make a decision at $s_t$.

The evaluation of each approach was conducted five times for the evaluation SOLAR set. The results, summarized in Figure 6a, show the success rates for: 1) Whether the agent reaches the correct answer state and 2) Whether the agent executes `Submit` operation in the answer state. When using the VAE prior, the agent reaches the correct answer state in only about 10% of test episodes and submits the answer in just 1%. With latents sampled using DDPM, about 10% of the answers are correctly submitted, while the agent reaches the answer state approximately 37% of the time. When using LDCQ, the agent reaches the answer state in over 90% of cases and successfully submits the correct answer in about 77% of test episodes. These results demonstrate that the Q-function enhances the agent's ability to both reach the correct answer and recognize when it has arrived at the answer state.

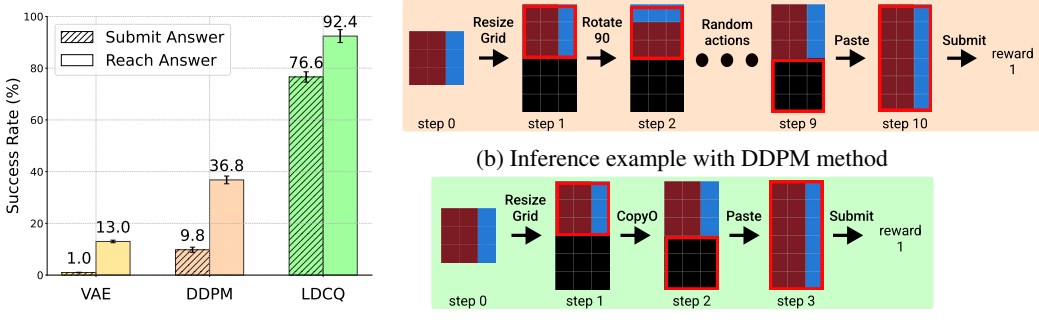

(a) Test accuracy for three methods

(b) Inference example with DDPM method

(c) Inference example with LDCQ method

Figure 6: (a) The evaluation results for 100 test examples for a simple task. LDCQ shows significantly improved performance compared to the other two methods, successfully reaching the correct answer state and executing the `Submit` operation at the answer state. The error bars represent the 96% confidence interval. (b) With the latent sampled with DDPM, the agent sometimes reaches the correct answer after performing various actions. This occurred rarely during evaluation, and even when it did, it did not appear in subsequent evaluations. (c) When using LDCQ, it often shows the case that skips unnecessary actions. The inference example with the VAE prior method is omitted because it rarely solves the problem.

Figure 6b and Figure 6c highlight the different solving strategies exhibited by the Q-function. When using the latent sampled with DDPM, the agent performs diverse actions, occasionally reaching the goal by chance. In contrast, with the Q-function, the agent consistently reaches the correct answer in every evaluation. In scenarios where the input grid is vertically symmetrical, the agent even skips unnecessary operation `FlipV` and proceeds directly to `Submit`. Notably, the training dataset does not include any trajectories where the `FlipV` operation is skipped, even for symmetrical grids. With the Q-function, the model recognizes that applying `FlipV` does not alter the state. Consequently, the Q-value for submitting at that state increases, prompting the agent to choose the `Submit` operation. This demonstrates the reasoning ability of the agent trained with LDCQ in solving ARC tasks, as recognizing when the correct answer state has been reached is crucial.

## 6 LIMITATIONS & DISCUSSIONS

In our experiment, the LDCQ method showed significant improvement in reaching the goal. However, in approximately 16% of cases, the agent reached the correct state but proceeded with another action instead of submitting the solution, even with the assistance of the Q-function. This issue arises because the Q-function, while enhancing decision-making, sometimes assigns higher values to actions other than submission, causing the agent to bypass the goal state. This suggests that the Q-function is not perfectly aligned with the final objective in ARC. Notably, in ARC tasks, even when solving different test examples within the same task where the same rule is applied, the actual action sequence can vary depending on factors like grid size or the arrangement of elements in the input grid. The current Q-values are calculated based on the absolute state values, which occasionally leads to misjudgments when submitting the correct solution. Therefore, improving the agent's ability to accurately determine when to submit the correct answer is necessary for future research.

While the LDCQ approach performs well in a simple ARC task setting, more complex tasks and multi-task environments present additional challenges. Unlike single-task scenarios, where the agent follows a fixed strategy toward a predefined answer, multi-task settings demand flexibility to adapt to changing goals or new possibilities during task execution. We expect that addressing these challenges could involve integrating task classifiers for Q-learning. Additionally, incorporating modules so that the agent can revise its strategy during task execution—adjusting based on evolving states or objectives rather than rigidly following the initial strategy—may enhance its adaptability.

In traditional supervised RL approaches, such as those described by Ghugare et al. (2024), stitching typically occurs only when the goal remains consistent across tasks. To address this limitation, we employed temporal data augmentation, which involves starting from an intermediate state near the goal and setting a new target. In SOLAR, this could be extended by using non-optimal paths as goals in non-optimal trajectories. However, in ARC, where goals are determined by demonstration pairs, augmenting all goals is impractical. More careful strategies are needed to enable stitching for entirely new goals not previously encountered. If methodologies are developed that can combine existing actions toward different goals, we expect that SOLAR will facilitate these combinations.

Going forward, refining how the Q-function evaluates states and actions will be crucial. To improve performance, especially in multi-task environments, incorporating mechanisms that not only assess the state and action in relation to the goal but also guide the agent toward the most effective path to achieve the ultimate objective will be beneficial. Recognizing the task's context and how close states are to the correct solution is essential for ensuring that the Q-function helps navigate toward the goal efficiently.

## 7 CONCLUSION

This research demonstrates the potential of offline reinforcement learning (RL), particularly the Latent Diffusion-Constrained Q-learning (LDCQ) method, for efficiently sequencing and organizing actions to solve tasks in grid-based environments like the Abstraction and Reasoning Corpus (ARC). This work is the first to tackle ARC using a diffusion-based offline RL model within a properly designed environment, guiding agents step-by-step toward correct solutions without generating the full ARC grid at once. Through training on SOLAR, we successfully applied and evaluated offline RL methods, showing that agents can learn to find paths to the correct answer state and recognize when they've reached it. This suggests that RL with a well-designed environment is promising for abductive reasoning problems, potentially reducing data dependency compared to traditional methods. As tasks become more complex, especially in multi-task settings, refining the Q-function to address unique reward structures is crucial, with multi-task environments requiring task-specific adaptations to account for varying states and rewards. Integrating modules like task classifiers or object detectors could enhance the agent's ability to dynamically adjust its strategy, promoting more flexible decision-making. This research opens new avenues for program synthesis in analogical reasoning tasks with RL environments, potentially integrating with analogy findings techniques (hypothesis search with LLMs) to handle a wider range of ARC tasks.

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

# A   TRAINING DETAILS

## A.1   LATENT DIFFUSION CONSTRAINED Q-LEARNING (LDCQ)

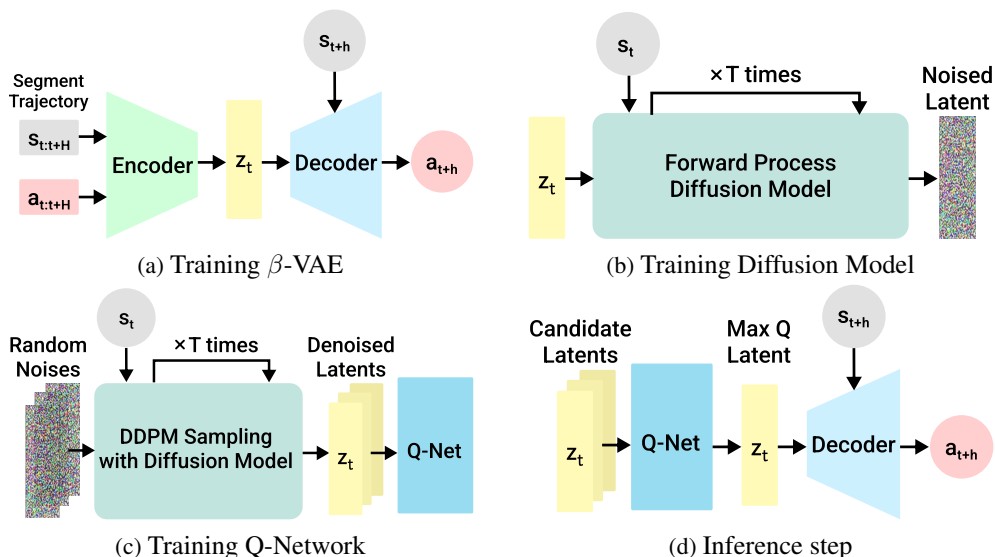

(a) Training $\beta$-VAE

(b) Training Diffusion Model

(c) Training Q-Network

(d) Inference step

Figure 7: (a)–(c) Training stages of LDCQ. (a) Training a $\beta$-VAE with an encoder that encodes $H$-horizon segment trajectories into latents $z_t$, and a policy decoder that decodes actions based on $z_t$ and state $s_{t+h}$ where $h \in [0, H)$ contained in the latent. (b) Training a diffusion model based on $z_t$ and the $s_t$. (c) Training a Q-network using latents sampled through the diffusion model. (d) LDCQ inference step at $s_{t+h}$. Possible latents at $s_t$ are sampled through the diffusion model, and the agent executes actions resulting from decoding the latent with the highest Q-value.

Latent Diffusion-Constrained Q-learning (LDCQ) (Venkatraman et al., 2024) leverages latent diffusion and batch-constrained Q-learning to handle long-horizon, sparse reward tasks more effectively. The LDCQ method uses sampled latents that encode trajectories of length $H$ to train the Q-function, effectively reducing extrapolation error. The training process of LDCQ is shown in Figure 7: 1) training the $\beta$-VAE to learn latent representations, 2) training the diffusion model using the latent vectors encoded by the $\beta$-VAE, and 3) training the Q-network with latents sampled from the diffusion model.

**Training Latent Encoder and Policy Decoder**   The first stage in training with LDCQ is to train a $\beta$-VAE that learns latent representations. In this stage, the $\beta$-VAE learns how actions are executed over multiple steps to change the state. With $H$-horizon latents, it becomes easier to capture longer-term changes in the state. We use SOLAR as the training dataset $\mathcal{D}$, which contains $H$-length segmented trajectories $\tau_t$. Each $\tau_t$ consists of state sequences $s_{t:t+H} = [s_t, s_{t+1}, ..., s_{t+H-1}]$ and action sequences $a_{t:t+H} = [a_t, a_{t+1}, ..., a_{t+H-1}]$, along with additional information such as demonstration examples. As shown in Figure 7a, during the $\beta$-VAE training stage, the encoder $q_\phi$ is trained to encode $\tau_t$ into the latent representation $z_t$, and the low-level policy decoder $\pi_\theta$ is trained to decode actions based on the given state and latent. For example, given the latent $z_t$ and a state from the segment trajectory, $s_{t+h}$ where $h \in [0, H)$, the policy decoder decodes the action $a_{t+h}$ for $s_{t+h}$. The $\beta$-VAE is trained by maximizing the evidence lower bound (ELBO), minimizing the loss in Eq. 1. The loss consists of the reconstruction loss from the low-level policy decoder and the KL divergence between the approximate posterior $q_\phi(z_t|\tau_t)$ and the prior $p_\omega(z_t|s_t)$.

$$\mathcal{L}_{\text{VAE}}(\theta, \phi, \omega) = -\mathbb{E}_{\tau_t \sim \mathcal{D}} \left[ \mathbb{E}_{q_\phi(z_t|\tau_t)} \left[ \sum_{l=t}^{t+H-1} \log \pi_\theta(a_l|s_l, z_t) \right] - \beta D_{KL}(q_\phi(z_t|\tau_t) \parallel p_\omega(z_t|s_t)) \right] \quad (1)$$

**Training Latent Diffusion Model**    In the second stage, latent diffusion model is trained to generate latents based on the latent representations encoded by the $\beta$-VAE. The training data consists of $(s_t, z_t)$ pairs, which are used to train a conditional latent diffusion model $p_\psi(z_t|s_t)$ by learning the denoising function $\mu_\psi(z_t^j, s_t, j)$, where $j \in [0, T]$ is diffusion timestep. This allows the model to capture the distribution of trajectory latents conditioned on $s_t$. $q(z_t^j|z_t^0)$ denotes the forward Gaussian diffusion process that noising the original data. Following previous research (Ramesh et al., 2022; Venkatraman et al., 2024), we predict the original latent rather than the noise, balancing the loss across diffusion timesteps using the Min-SNR-$\gamma$ strategy (Hang et al., 2023). The loss function used to train the diffusion model is shown in Eq. 2. Here, $z_t^j$, $j \in [0, T]$ represents noised latent on $j$-th diffusion time step, when $j = 0$ then $z_t^0 = z_t$ and $z_t^T$ is Gaussian noise.

$$\mathcal{L}(\psi) = \mathbb{E}_{j\sim[1,T],\tau_H\sim\mathcal{D},z_t\sim q_\phi(z_t|\tau_t),z_t^j\sim q(z_t^j|z_t^0)}\left[\min\{\mathrm{SNR}(j),\gamma\}\|z_t^0 - \mu_\psi(z_t^j, s_t, j)\|^2\right] \quad (2)$$

**Training Q-Network**    Finally, the latent vectors sampled by the latent diffusion model are used for Q-learning. For sampling latents, we use the DDPM method (Ho et al., 2020). The trained diffusion model samples latents by denoising random noise using the starting state information $s_t$. We use the data consisting of $(s_t, z_t, r_{t:t+H}, s_{t+H})$ for training Q-Network, where $r_{t:t+H} = \sum_{l=t}^{t+H-1} \gamma^l r_l$ deontes the discounted sum of rewards. Here, DDPM sampling is used to sample $z_{t+H}$ for $s_{t+H}$. For Q-learning, we use Clipped Double Q-learning (Fujimoto et al., 2018) as shown in Eq. 3 with Prioritized Experience Replay buffer (Schaul, 2016) to improve learning stability and mitigate overestimation. The trained Q-network $Q(s_t, z_t)$ evaluates the expected return of performing various $H$-length actions, with $z_t$ sampled via DDPM based on $s_t$. This allows the network to efficiently calculate the value of actions over $H$-steps to estimate future returns. Additionally, since ARC tasks involve inferring analogies from demonstration pairs, the embedded representation of the demonstration pair, $p_{emb}$, is also used in the Q-function calculation.

$$Q(s_t, z_t, p_{emb}) \leftarrow \left(r_{t:t+H} + \gamma^H Q(s_{t+H}, \underset{z\sim p_\psi(z_{t+H}|s_{t+H})}{\mathrm{argmax}} Q(s_{t+H}, z, p_{emb}), p_{emb})\right) \quad (3)$$

## A.2    Hyperparameters

We used a horizon length of 5 for encoding skill latents, meaning the model plans and evaluates actions over a five-step lookahead.

We trained the diffusion model with 500 diffusion steps. If the number of diffusion steps is too small, it can lead to high variance in the sampling process, potentially causing errors during the decoding of operations or selections in ARCLE. To minimize these errors, we set the number of diffusion steps to 500, ensuring more accurate operation and selection decoding from the sampled latents.

We set the discount factor to 0.5 to ensure the model appropriately balances immediate and future rewards. Since the total steps required to reach the correct answer in ARCLE are usually fewer than 20, a high discount factor could cause the agent to struggle in distinguishing between submitting at the correct state and continuing with additional steps, which could lead to episode failure.

The hyperparameters that we used for training three stages of LDCQ are shown in Tables 1, 2 and 3.

Table 1: Hyperparameters for training $\beta$-VAE

| Parameter | Value |
|---|---|
| Learning rate | 5e-5 |
| Batch size | 128 |
| Epochs | 400 |
| Horizon ($H$) | 5 |
| Latent dimension (z) | 256 |
| KL loss ratio ($\beta$) | 0.1 |
| Hidden layer dimension | 512 |

Table 2: Hyperparameters for training latent diffusion model

| Parameter | Value |
|---|---|
| Learning rate | 1e-4 |
| Batch size | 32 |
| Epochs | 400 |
| Diffusion steps ($T$) | 500 |
| Variance schedule | linear |
| Sampling algorithm | DDPM |
| $\gamma$ (For Min-SNR-$\gamma$ weighing) | 5 |

Table 3: Hyperparameters for training DQN

| Parameter | Value |
|---|---|
| Learning rate | 5e-4 |
| Batch size | 128 |
| Discount factor ($\gamma$) | 0.5 |
| Target net update rate ($\rho$) | 0.995 |
| PER buffer $\alpha$ | 0.7 |
| PER buffer $\beta$ | Linearly increased from 0.3 to 1, Grows by 0.03 every 2000 steps |
| Diffusion samples for batch $\mathrm{argmax}$ | 100 |

## A.3 HARDWARE

We used an NVIDIA A100-SXM4-40GB GPU to train the model. Training the $\beta$-VAE took about 7 hours, while training the diffusion model and Q-network each took around 6 to 10 hours.

# B    DETAILS OF SOLAR-GENERATOR

## B.1    OPERATIONS IN SOLAR

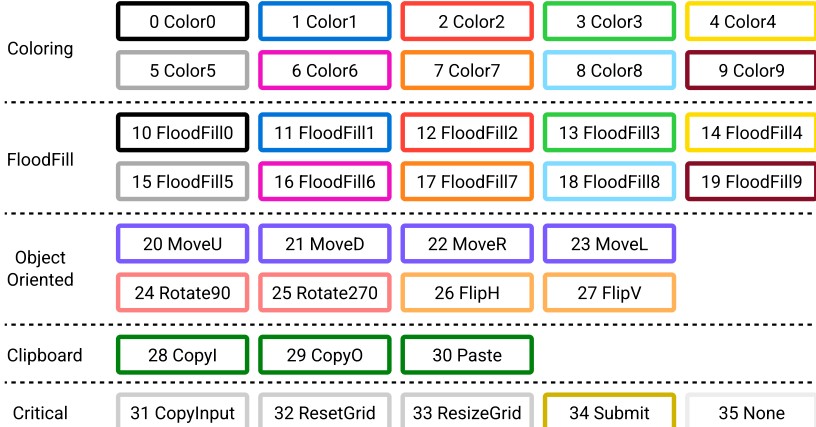

Figure 8: All operations compatible with SOLAR, 0–34 operations follow ARCLE, and only in SOLAR, 35 (`None`) is for terminated episode. It means the episode is ended after `Submit`.

The operations from 0 to 34 are identical to those used in ARCLE (Lee et al., 2024). Since `Submit` is an operation that receives a reward, it should only be used when the state is considered correct and not excessively. Due to LDCQ's fixed horizon, and to ensure that the agent only uses `Submit` when the state is definitely correct, we added a `None` operation that fills all subsequent states after `Submit` with the 11th color (10), which does not exist in the original ARC (0–9). In other words, during training, the `None` action emphasizes that the episode ends after `Submit`.

## B.2    GRID MAKER

For generating SOLAR, we create a SOLAR-Generator that can synthesize a large amount of data for a given rule. Grid Maker is a hard-coded program specific to each task. Grid Maker contains the rules for synthesizing demonstration examples and test examples, and the synthesized solution action path consists of operations and selections. In Grid Maker, data is formatted to be compatible with ARCLE. The Grid Maker constructs analogies with the same problem semantics but with various attributes such as the shape, color, size, and position of objects. SOLAR-Generator can generate intermediate trajectories by interacting with ARCLE. The algorithm of the SOLAR-Generator is designed to augment specific tasks using the Grid Maker, which can primarily be divided into three parts.

Grid Maker was built as a data loader, which is used in ARCLE. In the original ARCLE environment, there was no need to load operations and selections. Only the grid was loaded with original ARC. To change this structure, the entire environment would need to be recreated. Instead, operations and selections are now loaded from the data loader's description, allowing us to retain the original environment. Therefore, the process of creating input-output examples and generating action sequences works within a single file.

**Specifying Common Parts**    Each task in the ARC dataset usually contains 3 demonstration examples, with common elements observed across these pairs. In the common parts, attributes such as color, the type of task, and the presence of objects are predetermined using random values before pair generation.

**Synthesizing Examples**    In the example synthesis phase, the input of the original task is augmented in a way that ensures diversity while preserving the integrity of the problem-solving method. A random input grid is generated under conditions that satisfy the analogy required by the task. A solution grid is created using a hard-coded algorithm. For tasks involving pattern-based problems, as experimented in the paper, selections are made to fit the grid size, and various operations are executed

either randomly or in a predetermined order. For object-based problems, the solution grid is generated by an algorithm that finds the necessary objects in the input grid and processes them according to the task requirements.

**Converting to ARCLE Trajectories**   This stage involves the creation of an ARCLE-based trajectory that meticulously adheres to the problem-solving schema of the synthesized examples. The entire process is carried out through a hard-coded algorithm. During the example synthesis process, the locations of objects may already be known, or they can be identified using a search algorithm. The information obtained is then used to make the appropriate selections, and the trajectory is converted into an ARCLE trajectory through an algorithm that leads to the correct solution.

If all steps are properly coded, it is possible to generate the operations and selections that lead to the correct solution for any random input grid. These are then fed into ARCLE to obtain intermediate states, rewards, and other information, and to verify whether the correct result is reached. Once steps 1) to 3) are correctly implemented, SOLAR-Generator can continuously and automatically generate as much data for the given task as the user desires, using the Grid Maker.

### B.3   EXAMPLE OF DATA SYNTHESIS IN GRID MAKER AND THE GENERATION OF SOLAR

SOLAR-Generator can synthesize SOLAR for object-based tasks. Figure 9 shows a variant of Task 2 from Figure 1. Grid Maker generates random input grids with some variances first. In this variant, each episode randomly selects two colors for the boxes. Each inputs can have different grid sizes, and rules are established for objects of each color within the episode. Then it generates the answer output grids for the input grids through algorithm. The solution algorithm in Grid Maker proceeds as follows: 1) Find the top-left corner of the orange square and repeat the coloring process to draw a diagonal line to the grid's edge. 2) Find the bottom-right corner of the red square and repeatedly color diagonally until the end of the grid is reached. With these algorithms, Grid Maker can synthesize as many examples and SOLAR trajectories as the user desires.

Task 2_gold-standard_7

Demonstration Examples

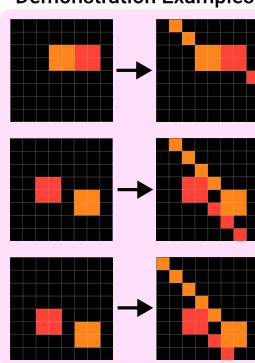

Test Example with Trajectory

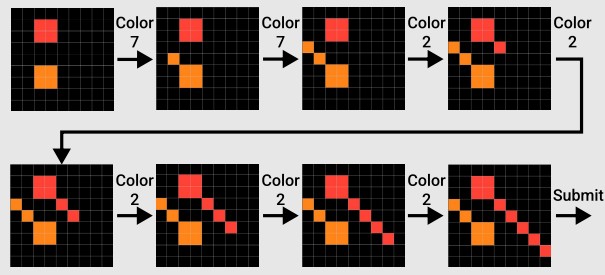

Figure 9: A gold standard trajectory for Task 2 in Figure 1. SOLAR contains its trajectory ID, demonstration examples, and a test example with trajectory.

### B.4 OTHER SOLAR EXAMPLES

Figure 10 illustrates two examples of episodes from the tasks used in the experiment. Each episode includes three random demonstration examples and a trajectory for a test example. Figure 10a shows a gold standard trajectory, which represents the ideal sequence of actions to reach the correct answer state. Figure 10b shows a non-optimal trajectory that, while not a gold standard, also reaches the answer state. The clip grid, reward, and termination information are not displayed.

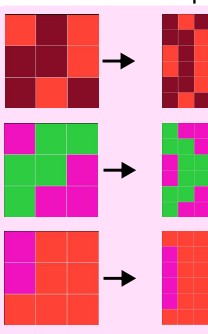

(a) Gold standard episode

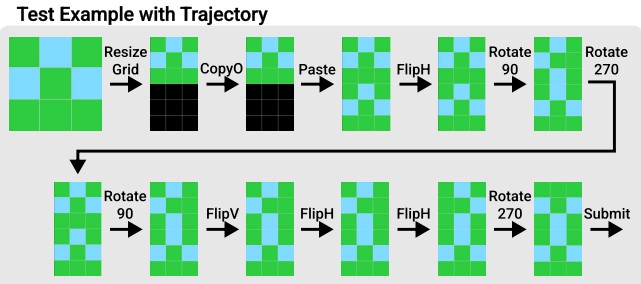

(b) Non-optimal episode

Figure 10: SOLAR episode examples of the task used in our experiment. Each episode contains three demonstration examples and a test example with a trajectory. (a) An example of a gold standard episode that ideally reaches the answer. (b) An example of a non-optimal episode that is not ideal, but still reaches the answer state.

**Task 1_gold-standard**

**Demonstration Examples**

**Test Example with Trajectory 1**

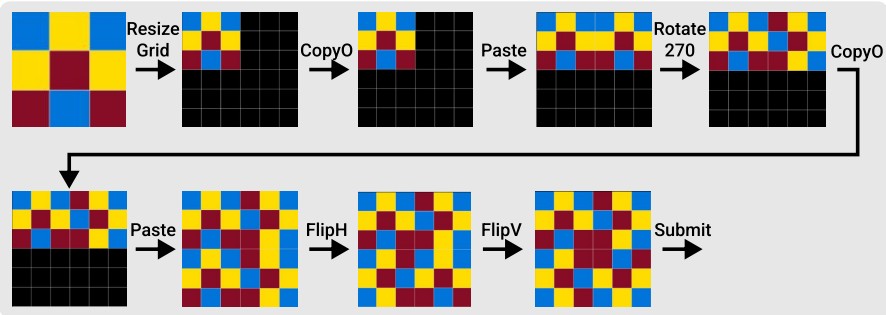

**Test Example with Trajectory 2**

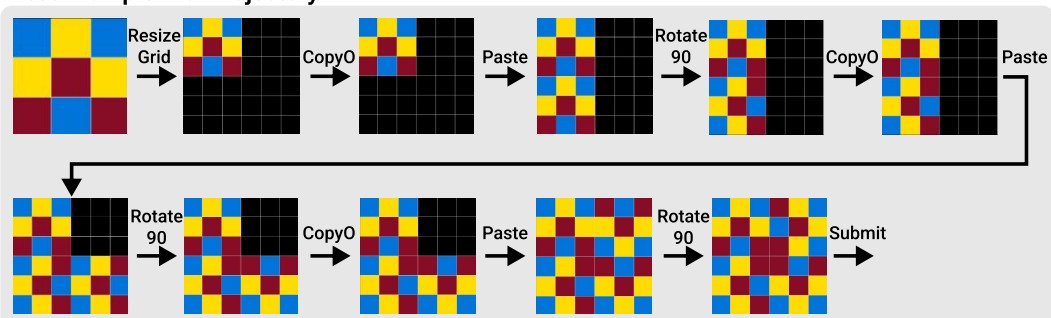

Figure 11: Two different gold standard trajectories for Task 1 in Figure 1, there might be multiple gold standard trajectories in the same task.

## C    RESULTS FOR MORE COMPLEX TASKS

We evaluate the agent by training with LDCQ in more complex tasks, Task 2 and Task 4.

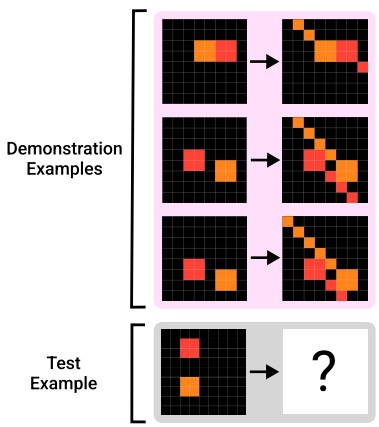
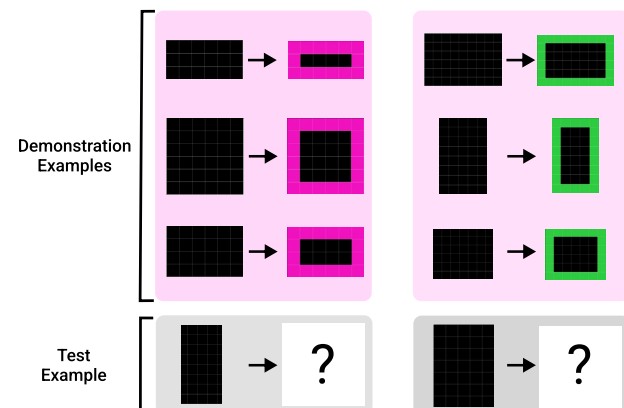

(a) An example of Task 2. It follows the rule of coloring based on the orange square and red square.

(b) An example of Task 4. The demonstration example pair in an episode uses the same color, while other episodes use colors ranging from 0 to 9.

Figure 12: Examples of Task 2 and Task 4.

Table 4: Accuracy for complex tasks. Each approach evaluated five times on the SOLAR evaluation set for each task. LDCQ shows improved performance compared to the other two methods, successfully reaching the correct answer and executing the `Submit` operation at the answer state.

| Task | Submit Answer | | | Reach Answer | | |
|---|---|---|---|---|---|---|
| | VAE | DDPM | LDCQ | VAE | DDPM | LDCQ |
| Task 2 | 0 | 0 | 1.2 | 5.2 | 3.4 | 9.8 |
| Task 4 | 0.4 | 0.2 | 4.0 | 0.4 | 0.2 | 4.0 |

Figure 12a illustrates Task 2, where the rule from the demonstration example is applied to solve the test example. In this task, since the coloring proceeds one cell at a time, each action's meaning depends on the object's position rather than being independently determined. Non-optimal trajectories were constructed by selecting locations slightly off the exact diagonal. The agent struggles to accurately select positions for the actions and to recognize when the correct state is reached. Although VAE and DDPM show some success in reaching the correct answer, there are no successful cases where the agent submits the answer grid. This suggests that the Q-function is crucial for correctly recognizing the solution state.

Figure 12b illustrates Task 4, which emphasizes recognizing and utilizing the colors used in the demonstration examples. Each episode in this task includes three demonstration examples, all of which share a common color. The agent needs to determine which color to use based on the colors in the demonstration examples. Non-optimal trajectories were generated by maintaining the selected rectangular region while choosing random colors. The agent struggled to select the correct color and had difficulty recovering once an incorrect color was chosen.

Although neither of these tasks achieved the high performance seen with the simple task described in the main text, using the Q-function still resulted in better performance compared to not using it. Since these tasks are more challenging than the simple task, with increased complexity in operations and selections, the limited training data might have contributed to the lower performance. Adjusting aspects like the discount factor in the Q-function training could also be beneficial. In future research, it would be worthwhile to investigate whether the challenges can be addressed by increasing the amount of data or if these tasks are fundamentally difficult for reinforcement learning to solve.

## D  SOLAR VS RE-ARC: COMPARING ARC DATA AUGMENTATION APPROACHES

During the development of SOLAR, another augmentation scheme called RE-ARC (Hodel, 2024) was independently developed and released. While both aim to generate diverse examples for ARC tasks, they differ significantly in their design objectives, underlying architectures, and dataset structures.

**Underlying Architectures**    SOLAR is built upon the ARCLE framework, designed for training reinforcement learning agents. It uses a limited set of actions based on the O2ARC web interface, which, despite their simplicity, are sufficient primitives to solve all ARC tasks. This design choice results in sequential trajectories directly applicable to reinforcement learning models. In contrast, RE-ARC is based on a more comprehensive Domain Specific Language (DSL) developed by Hodel, featuring 141 primitives. This expanded set of operations provides greater flexibility in expressing solutions, allowing for more complex transformations.

**Data Generation Approach**    SOLAR generates sequential trajectories that mirror the step-by-step approach humans use when solving ARC tasks. This aligns well with typical reinforcement learning models that execute actions sequentially. RE-ARC, leveraging its expansive DSL, generates solutions in the form of directed acyclic graphs (DAGs). This approach allows for more complex problem-solving strategies but may present challenges when applied to traditional reinforcement learning frameworks.

**Dataset Structure and Utility**    SOLAR provides complete episodes with detailed trajectories, including all intermediate states. This feature is crucial for training agents with offline reinforcement learning methods, allowing models to learn from the entire problem-solving process. RE-ARC focuses on augmenting input-output pairs without explicitly providing intermediate steps. While valuable for increasing example diversity and testing generalization capabilities, it may require additional processing for direct application in a reinforcement learning context.

**Flexibility and Potential for Integration**    SOLAR's design allows for easy generation of large episode datasets and is highly adaptable for various experimental setups in reinforcement learning research. The simplicity of the ARCLE action set makes it easier to modify and extend the system. RE-ARC's DAG-based approach, while not immediately compatible with sequential RL methods, opens up possibilities for more advanced RL frameworks capable of handling DAG-structured data.

**Future Directions**    Future research could explore synergies between SOLAR and RE-ARC approaches, potentially leading to more powerful and flexible AI systems for solving ARC tasks. One promising direction would be adapting the LDCQ methodology to work with RE-ARC's DAG structures, which could involve developing new RL algorithms capable of processing DAG-structured data. Another interesting avenue would be to investigate how SOLAR's sequential trajectories could inform or constrain the generation of more complex DAG-based solutions in RE-ARC. Such a hybrid approach could combine the simplicity and learnability of sequential actions with the expressiveness of DAG-based representations. By integrating the strengths of both approaches - SOLAR's alignment with current RL techniques and RE-ARC's comprehensive problem representations - we could potentially unlock new capabilities in AI systems. This integration might lead to significant advancements in abstract reasoning and problem-solving, bridging the gap between the efficiency of reinforcement learning and the expressiveness of symbolic methods.

