# OpenReview forum: "Diffusion-Based Offline RL for Improved Decision-Making in Augmented Single ARC Task"
_ICLR.cc/2025/Conference — Submitted to ICLR 2025_

### Official Review · Reviewer_xRWd · 2024-10-23

**Soundness:** 3
**Presentation:** 3
**Contribution:** 3
**Rating:** 5
**Confidence:** 4

**Summary:**

The paper presents a novel approach to addressing the challenges of strategic reasoning in abstract problem-solving using Latent Diffusion-Constrained Q-learning (LDCQ). The authors apply LDCQ, a prominent diffusion-based offline RL technique, to tasks from the Abstraction and Reasoning Corpus (ARC). Due to the limitations of the ARC dataset, they introduce Synthesized Offline Learning Data for Abstraction and Reasoning (SOLAR), which augments the original dataset with diverse trajectories for training offline RL agents. The paper shows that SOLAR, combined with LDCQ, enhances the agent’s ability to make multi-step sequential decisions and solve complex tasks that require abstract reasoning, highlighting the potential of diffusion-based offline RL to improve AI decision-making in environments requiring strategic and multi-step reasoning.

**Strengths:**

**Originality**: The application of diffusion-based offline RL to reasoning tasks, specifically through the introduction of LDCQ and the SOLAR dataset, is innovative and could open new avenues in RL for reasoning tasks.

**Quality**: The experiments are well-designed and support the paper’s claims. The comparisons between different approaches, including VAE and diffusion prior, are valuable.

**Clarity**: The structure and flow of the paper are generally clear, with useful visual aids and explanations of the ARC tasks and offline RL methodology.

**Significance**: If extended to more complex reasoning tasks, the proposed method could significantly improve how RL systems approach long-horizon decision-making problems, contributing to the broader field of AI reasoning.

**Weaknesses:**

**Data Augmentation Concerns**: While the SOLAR-Generator effectively augments training data, its reliance on hard-coded rules for action selection (Appendix Section B; specifically, Ln. 678-715)  might limit its applicability when tackling tasks that demand more dynamic or adaptive strategies. It'd be helpful for the authors to add more discussion about the limitations of the SOLAR-Generator's augmentation methods, and how it impacts scalability, transferability to other environments.

**Limited Comparisons**: The paper lacks sufficient comparative analysis with other offline RL approaches or state-of-the-art methods for ARC, making it difficult to gauge the improvement over prior methods/approaches quantitatively. Concretely, including and juxtaposing against any of the approaches from the ARCLE paper - for e.g.,  any sequential policy architecture  (like RNN policy of Vinyals et al. 2019) would greatly bolster the experimental results.

**Dataset Relevance**: SOLAR, while beneficial, is specifically tailored to ARC tasks (and the ARCLE environment), which may not generalize to other reasoning tasks or RL benchmarks (e.g. Meta-World, RLBench, CALVIN, NLE etc.). Extending the applicability of SOLAR to other domains would increase the paper’s impact.

**Questions:**

**Questions**

1. Data Augmentation Limitations: Could the authors elaborate more on how the deterministic nature of the SOLAR-Generator might affect generalization in more complex or varied task environments?

2. Could the authors provide more quantitative comparisons between LDCQ and other reinforcement learning techniques, especially in terms of training efficiency and performance?

3. Can the proposed LDCQ approach be generalized to other abstract reasoning benchmarks outside of ARC / ARCLE? What are the main challenges in doing so?

4. The LDCQ method relies on generating latent samples through diffusion models. Could the authors elaborate on the computational cost of this approach and how it scales with more complex tasks?

---

> ### Author Response · Authors · 2024-11-26
> **Response to Reviewer xRWd**
>
> **Weakness 1, 3 & Q3.**
> It is true that relying on hard-coded rules to generate gold-standard trajectories makes augmentation more challenging for complex tasks. However, SOLAR includes augmented data for approximately 20 tasks, some of which are relatively complex. One of the primary goals of SOLAR is to ensure that the tasks learned by the model can be solved perfectly. Therefore, we expect that training with trajectories generated by SOLAR will demonstrate significant value in this regard. For simple tasks, we compared 100 test dataset problems with 500 training dataset problems and found no duplicates, which indicates that the model trained with SOLAR can solve new problems even within the same task.
>
> While the augmentation process is not automated, solving ARC tasks inherently involves diverse and complex augmentation methods. ARC is a benchmark that inherently requires core knowledge priors and reasoning abilities to solve a diverse range of tasks. Solving ARC tasks in itself provides significant value, so we did not emphasize extending to environments beyond ARC. Rather than focusing on extending SOLAR itself to other benchmarks, we believe that advancing methodologies using SOLAR to solve ARC tasks could serve as a key approach for tackling other complex reasoning tasks in the future.
>
> **Weakness 2 & Q2.**
> In this paper, LDCQ[1] was used as an example of an offline RL method to demonstrate that SOLAR can be used for learning through such offline RL approaches. Therefore, we did not conduct separate experiments on other baselines, such as non-diffusion policies. Instead, to demonstrate the effectiveness of the Q-function, we compared the sampling method (VAE prior, DDPM) using the VAE and Latent Diffusion Model used in LDCQ training with the RL method (LDCQ). As you mentioned, it would be worthwhile to compare with other methods. Since the main contribution is the introduction of SOLAR and demonstrating its applicability to offline RL, we have revised the main text to give greater emphasis to SOLAR's contribution. The revised sections are highlighted in blue, and we would appreciate it if you could review them.
>
> **Q1.**
> We conducted additional experiments on two tasks beyond those used in the paper: 1) a task that performs different rules depending on the colors of two squares, and 2) a task where the edges of the grid are colored based on the given problem (Appendix C). The results showed that our method did not achieve high performance on either task. However, compared to the sampling method such as VAE, DDPM, using the Q-function as guidance led to better performance. Therefore, as mentioned in the Section 6 limitation, we believe that if the Q-function is improved, it could demonstrate reasoning abilities. Furthermore, solving ARC tasks in itself provides significant value, so we did not emphasize extending to environments beyond ARC.
>
> **Q4.**
> Appendix A.3 provides information about the hardware used and the approximate time required for training. The amount of data required for training also varies significantly by task, and computational costs will differ accordingly. The variation in data requirements per task is a major challenge and may be a limitation of the current reinforcement learning methodologies. In future research, it may be useful to analyze the amount of data needed for each task, or consider a methodological shift to make the required data smaller and more consistent across tasks.
>
> [1] Siddarth Venkatraman, Shivesh Khaitan, Ravi Tej Akella, John Dolan, Jeff Schneider, and Glen Berseth. Reasoning with Latent Diffusion in Offline Reinforcement Learning. In ICLR, 2024.

---

> > ### Comment · Reviewer_xRWd · 2024-11-26
> > **Reply to Author Responses**
> >
> > Dear Authors,
> >
> > Thank you for your responses and manuscript updates.
> >
> > Upon reviewing them and along with the other reviews as a whole, **I am deciding to stay with my original assessment**.
> >
> > Overall, the work is a validation of something well-established (abstractly, data augmentation + caveats work (including diffusion-based paradigm)) for a particular domain. However, the hard-coded, domain specific applicability (albeit complex) of SOLAR leaves me unconvinced of its overall impact and adoption for/by the research community.

---

### Official Review · Reviewer_bXYY · 2024-10-31

**Soundness:** 2
**Presentation:** 2
**Contribution:** 2
**Rating:** 3
**Confidence:** 4

**Summary:**

The paper addresses the shortage of experience data necessary for training offline reinforcement learning (RL) methods on Abstraction and Reasoning Corpus (ARC) tasks. The motivation behind this work is to leverage ARC tasks to assess the reasoning capabilities of offline RL agents that incorporate diffusion models. In particular, the authors focus on evaluating the performance of Latent Diffusion-Constrained Q-learning (LDCQ). They introduced the Synthesized Offline Learning Data for Abstraction and Reasoning (SOLAR) dataset, generated by the SOLAR-Generator, which provides diverse trajectory data to ensure sufficient training experience for the learning agent. Experimental results show that the LDCQ method can correctly make multi-step sequential decisions to complete ARC tasks, thereby demonstrating reasoning capabilities.

**Strengths:**

- The paper is clear, well-organized, and visually informative.
- ARC tasks are challenging and suitable for demonstrating the strengths of RL methods.
- The LDCQ agent’s ability to omit unnecessary actions from the original data is notable (lines 424-430).

**Weaknesses:**

- The focus solely on evaluating diffusion-based offline RL, particularly LDCQ, seems narrow. Comparing traditional offline RL methods like CQL [1], IQL [2], or EDAC [3], as well as generative-guided agents such as Decision Transformers (DT) [4], would add depth and increase the significance of the work.
- The experiment is limited to a simple ARC setting. More complex ARC tasks should be investigated to create a major contribution.
- The paper’s main contribution is unclear. It’s uncertain whether the primary aim is to introduce an augmented dataset or to assess diffusion-model-based offline RL agents.
   + If introducing a dataset is the goal, the focus could shift to complex or multi-task settings, and testing with diverse RL methods to show the usefulness of the dataset. Additionally, a more thorough introduction to ARC tasks would be beneficial.
   + If evaluating diffusion-based RL is the goal, it would be beneficial to include additional diffusion-based algorithms, like Synther [5], and test on more complex ARC tasks or in multi-task scenarios.


[1] Kumar, A., Zhou, A., Tucker, G., & Levine, S. (2020). Conservative q-learning for offline reinforcement learning. Advances in Neural Information Processing Systems, 33, 1179-1191.

[2] Kostrikov, I., Nair, A., & Levine, S. (2021). Offline reinforcement learning with implicit q-learning. arXiv preprint arXiv:2110.06169.

[3] An, G., Moon, S., Kim, J. H., & Song, H. O. (2021). Uncertainty-based offline reinforcement learning with diversified q-ensemble. Advances in neural information processing systems, 34, 7436-7447.

[4] Chen, L., Lu, K., Rajeswaran, A., Lee, K., Grover, A., Laskin, M., ... & Mordatch, I. (2021). Decision transformer: Reinforcement learning via sequence modeling. Advances in neural information processing systems, 34, 15084-15097.

[5] Lu, C., Ball, P., Teh, Y. W., & Parker-Holder, J. (2024). Synthetic experience replay. Advances in Neural Information Processing Systems, 36.

**Questions:**

Why didn’t the authors use the dataset to evaluate other offline RL approaches? It’s unclear why LDCQ would be uniquely suited to the generated data.

---

> ### Author Response · Authors · 2024-11-26
> **Response to Reviewer bXYY**
>
> **Weakness 1 & Q1.**
> In this paper, LDCQ[1] was used as an example of an offline RL method to demonstrate that SOLAR can be used for learning through such offline RL approaches. Therefore, we did not conduct separate experiments on other baselines, such as non-diffusion policies. Instead, to demonstrate the effectiveness of the Q-function, we compared the sampling method (VAE prior, DDPM) using the VAE and Latent Diffusion Model used in LDCQ training with the RL method (LDCQ). As you mentioned, it would be worthwhile to compare with other methods. Since the main contribution is the introduction of SOLAR and demonstrating its applicability to offline RL, we have revised the main text to give greater emphasis to SOLAR's contribution. The revised sections are highlighted in blue, and we would appreciate it if you could review them.
>
> **Weakness 2.**
> We conducted additional experiments on two tasks beyond those used in the paper: 1) a task that performs different rules depending on the colors of two squares, and 2) a task where the edges of the grid are colored based on the given problem (Appendix C). The results showed that our method did not achieve high performance on either task. However, compared to the sampling method such as VAE, DDPM, using the Q-function as guidance led to better performance. Therefore, as mentioned in the Section 6 limitation, we believe that if the Q-function is improved, it could demonstrate reasoning abilities.
>
> **Weakness 3.**
> In this study, instead of comparing LDCQ with existing offline reinforcement learning methods, we adapted it to fit the unique characteristics of ARC tasks. The main focus of this research is to demonstrate that offline reinforcement learning can solve these tasks through a structured combination of actions that reflects the reasoning process required to solve ARC tasks. While it is also important to demonstrate that a representative offline RL method can learn effectively, the main contribution of this study lies in introducing SOLAR and attempting to solve ARC using diffusion-based RL. Additional experimental results are as mentioned in weakness 2.
>
> [1] Siddarth Venkatraman, Shivesh Khaitan, Ravi Tej Akella, John Dolan, Jeff Schneider, and Glen Berseth. Reasoning with Latent Diffusion in Offline Reinforcement Learning. In ICLR, 2024.

---

### Official Review · Reviewer_fTpe · 2024-10-31

**Soundness:** 3
**Presentation:** 3
**Contribution:** 2
**Rating:** 5
**Confidence:** 3

**Summary:**

This paper examines the ARC reasoning tasks. In this setting, agents are provided with demonstration examples, and must correctly manipulate a new grid using the inferred strategy. This paper takes an RL view to this challenge, using the ARLET environment. The approach is to synthesize additional data and trajectories, then use these trajectories to train a policy. The SOLAR-Generator approach is to synthesize data following ground-truth patterns. These trajectories are then filtered to ensure they are valid trajectories, then labelled with standard RL information (reward, termination, etc). This uses a pre-defined Grid Maker helper.

**Strengths:**

This paper provides a clean, well-described approach to solving a challenging reasoning setting. All design choices are explained clearly, and figures provide a visual intuition as to what is going on. The proposed SOLAR-Generator is well-reasoned and suitably achieves the desired effect. While many decisions are domain-specific, they remain simple and are well justified.

**Weaknesses:**

A persistent weakness in this paper is the conflation of implementation details with the main contribution. For example, the paper spends quite a large portion describing the diffusion-based RL policy, when the key takeaway from the paper is the synthetic data which enables solving ARC tasks. The two non-LDCQ baselines shown in 5.2 are lacking any Q signal at all. A simple and more informative baseline would be to train a simple non-diffusion policy on the discrete space to maximize Q.

The section 3.2 on describing the SOLAR generation method could benefit from using less abstraction, and show some concrete examples of data generated.

As the paper only focuses on this single setting, its potential future contribution is limited.

Additionally, there are no clear comparisons to past methods, making it hard to judge the effect of the proposed strategies.

**Questions:**

- Why is it necessary to train a latent representation of the action space, given that the action space is small and discrete, and as mentioned in the paper, a typical horizon for solving the task is only 5 steps?
- It would strengthen the paper to focus specifically on the SOLAR generation, and provide ablations on how performance scales with data, etc. The choice of policy network and sampling strategy (LDCQ) appears arbitrary, and distracts from the main contribution.
- A slight confusion when reading may clear up for future readers -- at which point is the agent conditioned on the demonstration examples when evaluated on a new task?
- Figure 3 can benefit from describing how the Q-network is trained. The core information is that it is trained via TD learning, which is missing in the figure. Figure three (b) is also counterintuitive, why does the model output a noised latent?

---

> ### Author Response · Authors · 2024-11-26
> **Response to Reviewer 9gqa fTpe**
>
> **Weakness 1, 4.**
> In this paper, LDCQ [1] was used as an example of an offline RL method to demonstrate that SOLAR can be applied to offline RL. Instead of conducting experiments with other baselines, we adapted LDCQ to fit the unique characteristics of ARC tasks. To demonstrate the effectiveness of the Q-function, we compared sampling methods (VAE prior, DDPM) used in LDCQ training with the RL method (LDCQ).
>
> The main contribution of this study lies in introducing SOLAR and demonstrating its applicability to offline RL. We have revised the main text to place greater emphasis on SOLAR's contribution, with the revised sections highlighted in blue for your review. The primary focus of this research was to show that ARC tasks can be solved using offline RL through action combinations, enabling reasoning and problem-solving.
>
> **Weakness 2 & Q2.**
> To provide a more detailed explanation of the SOLAR generation method in Section 3.2, we have revised the sentences and added the algorithm for the SOLAR-Generator. The revised parts are highlighted in blue, and we kindly ask you to refer to Algorithm 1. More specific details and examples of the SOLAR generation process are extensively covered in Appendix B. Examples of the generated SOLAR can also be seen in Figure 4 of Section 4, which shows the SOLAR used in the experiments.
> And the detailed explanation of LDCQ that was disrupting the flow has been moved to Appendix A.
>
> **Weakness 3.**
> We conducted additional experiments on two tasks beyond those used in the paper: 1) a task that performs different rules depending on the colors of two squares, and 2) a task where the edges of the grid are colored based on the given problem (Appendix C). The results showed that our method did not achieve high performance on either task. However, compared to the sampling method such as VAE, DDPM, using the Q-function as guidance led to better performance. Therefore, as mentioned in the Section 6 limitation, we believe that if the Q-function is improved, it could demonstrate reasoning abilities.
>
> **Q1.**
> The advantage of LDCQ lies in its ability to compress multiple steps into a latent representation, predicting multiple steps at once. Based on ARCLE, most ARC tasks are solved within 5 to 10 steps, typically under 20, so we divided them into units of 5 steps—this is a tunable hyperparameter. In solving ARC, even for the same task, there are a wide variety of test examples, making it important to infer which stage of the problem-solving process the agent is in for unseen states during training. Therefore, latent representation can be beneficial. Additionally, while the ARCLE space may seem small due to its discrete nature, in practice, it is defined as a combination of operations and selections, making it more complex to predict than it initially appears.
>
> **Q3.**
> Currently, the Q-function is computed in the form Q(s, a \mid \text{embedding}), where demonstration examples are embedded and incorporated into the calculation. $\boldsymbol{p}_{emb}$ denotes pair embedding. (Modify Appendix A Eq.3)
>
> $$
> Q(s_t, z_{t}, p_{emb}) \leftarrow \left( r_{t:t+H} + \gamma^H Q\left(s_{t+H}, argmaxQ(s_{t+H}, z, p_{emb})\right), p_{emb} \right)
> $$
>
> This serves two purposes. First, it uses the demonstration examples to classify tasks while learning the Q-function. Second, the actions may vary depending on the demonstration examples. As shown in Figure 12 (b), it is not possible to infer the action based solely on the state. Demonstration examples are required to accurately infer the correct color, and the demonstration pair must be considered to select the correct coloring action.
>
> **Q4.**
> The explanation of LDCQ, as it is not a primary focus, has been moved to Appendix A. Figure 9 corresponds to the original Figure 3.
>
> Although using TD learning is an important aspect of our approach, we did not include a separate illustration for it, as it only involves a minor modification of existing methodologies. The primary focus of this study is to demonstrate that the proposed reinforcement learning approach can effectively learn the desired behavior, which is why we developed SOLAR. Our emphasis lies in showcasing the viability of reinforcement learning for solving challenging reasoning tasks, supported by the structured data provided by SOLAR.
>
> (b) illustrates the forward process of training the Diffusion Model. This follows the training process of the Diffusion Model used in the DDPM [2], where the model learns the forward process of adding noise to the original data. The denoising process, shown as (c) the backward process (DDPM sampling), is then used to generate data by reversing the added noise.
>
> [1] Siddarth Venkatraman, Shivesh Khaitan, Ravi Tej Akella, John Dolan, Jeff Schneider, and Glen Berseth. Reasoning with Latent Diffusion in Offline Reinforcement Learning. In ICLR, 2024.
>
> [2] Jonathan Ho, Ajay Jain, and Pieter Abbeel. Denoising Diffusion Probabilistic Models. In NeurIPS, 2020

---

### Official Review · Reviewer_9gqa · 2024-11-04

**Soundness:** 2
**Presentation:** 2
**Contribution:** 1
**Rating:** 3
**Confidence:** 3

**Summary:**

This paper develops an augmented offline RL dataset called SOLAR  (Synthesized Offline Learning Data for Abstraction and Reasoning), to mitigate insufficient data in Abstraction and Reasoning Corpus (ARC).

**Strengths:**

1. The SOLAR-Generator enables customization of augmented data by generating different trajectory data based on predefined rules.
2. The augmented dataset helps demonstrate the effectiveness of offline RL algorithms, particularly in enhancing reasoning abilities.

**Weaknesses:**

1. Limited novelty: This paper primarily focuses on evaluating existing methods rather than developing new algorithms or applications. Its main contribution is the generation of a synthesized dataset to augment limited data, which is commonly used in previous works, such as [A1] in reinforcement learning and [A2] in computer vision. The authors should clarify the unique contributions and specific challenges involved in designing the SOLAR-Generator for ARC.
2. Quality of Generated Data: The random generation of synthetic samples raises concerns about ensuring optimal or efficient trajectories. It's crucial to determine whether training data containing episodes that overshoot the target state could negatively impact the agent's performance.
3. Unclear level of Data Diversity: The diversity of the generated data is unclear. Experimental procedures suggest that data augmentation is based on the same input and target, with variations introduced at random steps. This approach may result in data that lacks sufficient variability, potentially limiting the model's ability to generalize.
4. Quantifying Improvement and Insufficient Experiments: The paper does not provide clear metrics or comparisons to baseline methods within the ARC framework, making it difficult to evaluate the actual benefits of using the augmented data. Including quantitative evaluations with various agents would enhance understanding of the effectiveness of the proposed approach.

[A1] Laskin, Misha, et al. "Reinforcement learning with augmented data." Advances in neural information processing systems 33 (2020): 19884-19895.

[A2] Yang, Suorong, et al. "Image data augmentation for deep learning: A survey." arXiv preprint arXiv:2204.08610 (2022).

**Questions:**

1. How does the SOLAR-Generator ensure the quality of the generated trajectories?
2. What specific performance improvements were observed when using the augmented dataset compared to standard training data?
3. To what extent does the generated data vary to provide meaningful diversity, and how does this diversity impact the model's generalization capabilities?

---

> ### Author Response · Authors · 2024-11-26
> **Response to Reviewer 9gqa (1/2)**
>
> **Weakness 1.**
> First, the augmentation methods used in [A1] and [A2] primarily involve simple transformations such as rotation, color changes, resizing, and cropping of the data. These methods are relatively straightforward compared to the approach employed in SOLAR. In the simpler tasks discussed in the paper, we applied similar methods—like adjusting size or color—which could also be considered augmentations. However, SOLAR goes a step further by augmenting not just a single task but various tasks, allowing for the creation of new states by modifying elements such as the number of objects or their arrangement within the entire state grid.
>
> In other words, SOLAR is capable of not only making simple transformations to existing states but also modifying the rules that apply across all demonstration and test examples for a given task. It even creates new states that adhere to these modified rules. For instance, as shown in Figure 9 of Appendix B3, which is based on Task 2 from Figure 1 in the Introduction, we modified the rules associated with each color and generated new positions for the 2x2 squares. Users can even adjust the size of these squares if needed. Therefore, this process generates new problems that share the same underlying rules, allowing for diverse problem generation within a single task. This enables AI systems to learn the analogy required by the task and assess whether the learned analogy can be effectively applied.
>
> In comparison, the RE-ARC dataset provides a method for augmenting input-output pairs. However, as highlighted in Appendix C, the key distinction of SOLAR lies in its ability to augment intermediate states during the solution process of ARC tasks. SOLAR incorporates transitions derived from reinforcement learning into the generated trajectories. For instance, as illustrated in Figure 9, the SOLAR-Generator can adjust object colors, rearrange them, and randomly redefine the meanings associated with each color.
>
> Each ARC task has specific rules that must be followed to solve it. To claim that an AI system has fully understood the rules of a task, it should not only solve a single instance but also demonstrate strong performance across a variety of problems that share those rules. Therefore, we believe that it is essential to train trajectories across diverse input-output pairs to evaluate understanding effectively. Ultimately, the goal of SOLAR is to provide a dataset that allows ARC tasks to be tackled using offline reinforcement learning methods, while also generating diverse problems to evaluate whether the rules of the task have been fully understood.
>
> **Weakness 2.**
> We understood 'overshoot' as a situation where the agent goes beyond the desired correct state.
>
> In ARCLE, rewards are only provided when the agent executes the $\texttt{Submit}$ action at a correct state. This ensures that any episode with overshoot is inherently treated as non-optimal. Gold-standard trajectories reach the correct state in the most efficient manner, while non-optimal trajectories may or may not reach the correct state. When generating gold-standard trajectories, the SOLAR-Generator verifies that the final state of the trajectory is indeed correct. In the ARCLE-based environment, an agent only receives rewards if it executes the $\texttt{Submit}$ action at the correct state. Therefore, episodes that overshoot the correct state are automatically classified as non-optimal.
>
> Even in situations where non-optimal episodes are mixed with gold-standard trajectories, our experiments demonstrate the agent’s ability to follow the gold-standard trajectory. As shown in the results in Figure 7(a), the metric for "reach answer" is higher than that for "submit answer." This indicates the presence of test episodes with overshoot, where the agent reaches the correct state but fails to execute the $\texttt{Submit}$ action at the appropriate step.

---

> ### Author Response · Authors · 2024-11-26
> **Response to Reviewer 9gqa (2/2)**
>
> **Weakness 3 & Q3.**
> Not all trajectories are generated from the same input-output pair. The task used in the experiment is named 'Simple task,' and the precise augmentation process for generating trajectories is as follows:
>
> 1. **Input-Output Pair Generation**: Grid Maker creates new input-output pairs according to the conditions of the given task. In this experiment, 500 test example input-output pairs were generated.
> 2. **Action Sequence Generation**: Grid Maker generates the action sequence required to solve the test example. Each task's Grid Maker has a hard-coded algorithm that calculates which operations and selections are needed to produce the output from the generated input. The resulting action sequences are referred to as 'gold-standard' because they solve the problems without any unnecessary actions. From a certain point in the gold-standard, random actions are added to create 'non-optimal' trajectories. In the Simple task used in the main experiment, eight random actions are taken from the branch point, followed by \texttt{Submit}. Through this process, a total of nine non-optimal action sequences are generated from each gold-standard action sequence. Most non-optimal sequences do not reach the correct answer, but some reach it by chance, contributing to the diversity of the solution trajectories.
> 3. **Dataset Composition**: Based on the generated input-output pairs and action sequences, the dataset is composed by obtaining information such as intermediate states, rewards, and termination states through ARCLE.
>
> Additionally, we checked for overlap between the evaluation set and training set test examples. In the Simple task, the evaluation set was composed entirely of new test examples, allowing us to verify that the model understood and applied the task rules even to previously unseen examples. This demonstrates the model's generalization capability. While the states in non-optimal sequences may not be necessary for the Simple task, they could be valuable for other tasks in a multi-task environment, potentially enhancing the model's generalization ability. We also modified the content of section 4 to better understand.
>
> **Weakness 4.**
> In this paper, LDCQ[1] was used as an example of an offline RL method to demonstrate that SOLAR can be used for learning through such offline RL approaches. Therefore, we did not conduct separate experiments on other baselines, such as non-diffusion policies. Instead, to demonstrate the effectiveness of the Q-function, we compared the sampling method (VAE prior, DDPM) using the VAE and Latent Diffusion Model used in LDCQ training with the RL method (LDCQ). As you mentioned, it would be worthwhile to compare with other methods. Since the main contribution is the introduction of SOLAR and demonstrating its applicability to offline RL, we have revised the main text to give greater emphasis to SOLAR's contribution. The revised sections are highlighted in blue, and we would appreciate it if you could review them.
>
> **Q1.**
> As mentioned in the response to Weakness 3, multiple non-optimal trajectories are generated based on a single gold-standard trajectory for each input-output pair. Each task's Grid Maker has a hard-coded algorithm that calculates the necessary operations and selections to produce the output for the given input. The trajectory generated through this algorithm is the gold-standard trajectory. Since gold-standard trajectories are verified to ensure they successfully reach the correct state, they always achieve the correct answer. In contrast, non-optimal trajectories are intentionally included to lower the overall quality of the dataset and increase the decision-making difficulty. We aimed to demonstrate the potential of reinforcement learning by finding actions that contribute to solving the problem within a dataset containing unnecessary actions from non-optimal trajectories.
>
> **Q2.**
> The original ARC training data includes several demonstration examples but only one test example per task, which is insufficient for training deep learning models effectively. We believe that a model that can solve a diverse set of test examples, all considered part of the same task, is truly capable of understanding and solving that task. Our goal is to train the model by providing sufficient data for tasks that share the same rules, thereby enabling it to fully understand and reason through the given task. Therefore, rather than focusing on performance improvements using an augmented dataset compared to the original ARC data, this study aims to highlight the importance of enabling sequential decision-making and reasoning through offline reinforcement learning, which would be difficult to achieve with the original ARC data.
>
> [1] Siddarth Venkatraman, Shivesh Khaitan, Ravi Tej Akella, John Dolan, Jeff Schneider, and Glen Berseth. Reasoning with Latent Diffusion in Offline Reinforcement Learning. In ICLR, 2024.

---

### Author Response · Authors · 2024-11-26
**General Response**

Dear Reviewers and Meta-Reviewers,

We sincerely appreciate all the insightful feedback provided. We include the responses in the revised manuscript. For ease of reviewing, we highlight the added or revised text in **blue color**.

Below, we summarize the key improvements made in response to your comments, with changes highlighted in red for easy reference.

- Clarified the main contributions of SOLAR, emphasizing its advanced data augmentation methods that go beyond simple transformations, moving experimental LDCQ details to Appendix A. (Reviewer 9gqa, Reviewer fTpe, Reviewer bXYY, Reviewer xRWd)
- Added additional experiments on two complex ARC tasks in Appendix C. (Reviewer fTpe , Reviewer bXYY, Reviewer xRWd)
- Clarified how demonstration examples are embedded into the Q-function computation in Appendix A. (Reviewer fTpe)

We appreciate your constructive feedback and welcome any further questions or suggestions on OpenReview. We will continue to address all concerns following the reviewing policy.

---

### Meta-Review · Area_Chair_xXkU · 2024-12-20

**Metareview:**

The paper proposes a data augmentation technique (SOLAR) for around 20 tasks on the Abstraction and Reasoning Corpus (ARC), so that offline RL techniques run on the data can show improved performance on ARC.
All of the reviewers agreed that the proposal is sound, however the significance is pretty limited. For the core contribution of a data augmentation technique, important ablations are missing (e.g. performance and costs as the dataset sizes change, performance as the offline RL technique is varied across SOTA approaches, etc.). Moreover, the reviewers were rightly doubtful about the transferability of the technique (e.g. beyond ARC, or even beyond the 20 tasks for which rules can be hand-coded) .

**Additional Comments On Reviewer Discussion:**

The authors added experiments on some more ARC tasks during the rebuttal, as well as clarified their core contributions. However the main weaknesses were not addressed. Conducting ablation experiments to study the benefit of SOLAR augmentation, and comparing the achieved benefits across many different RL techniques may strengthen the current contributions.

---

### Decision · Program_Chairs · 2025-01-22

Reject